# Plasma Markers for Therapy Response Monitoring in Patients with Neuroendocrine Tumors Undergoing Peptide Receptor Radionuclide Therapy

**DOI:** 10.3390/cancers15245717

**Published:** 2023-12-06

**Authors:** Christoph Wetz, Tristan Ruhwedel, Imke Schatka, Jane Grabowski, Henning Jann, Giulia Metzger, Markus Galler, Holger Amthauer, Julian M. M. Rogasch

**Affiliations:** 1Department of Nuclear Medicine, Charité—Universitätsmedizin Berlin, Augustenburger Platz 1, 13353 Berlin, Germany; tristan.ruhwedel@charite.de (T.R.); giulia.metzger@charite.de (G.M.); markus.galler@charite.de (M.G.); holger.amthauer@charite.de (H.A.); julian.rogasch@charite.de (J.M.M.R.); 2Berlin Institute of Health at Charité—Universitätsmedizin Berlin, 10117 Berlin, Germany; jane.grabowski@gmail.com; 3Department of Hepatology and Gastroenterology, Charité—Universitätsmedizin Berlin, Augustenburger Platz 1, 13353 Berlin, Germany; henning.jann@charite.de

**Keywords:** neuroendocrine tumor, PRRT, plasma marker alterations, ALP, CgA, De Ritis ratio, patient response

## Abstract

**Simple Summary:**

Peptide receptor radionuclide therapy is a well-established therapy for the treatment of neuroendocrine tumors. Therapy typically consists of four cycles administered in 8–10 week intervals, resulting in an average treatment length of 8–10 months. Given the extensive treatment length, early identification of patients with disease progression could help to optimize disease management. Blood-based biomarkers such as chromogranin A, alkaline phosphatase (ALP), or the De Ritis ratio (aspartate aminotransferase/alanine aminotransferase) are continually assessed during treatment and represent a noninvasive and easily accessible source for the intratherapeutic monitoring of patients. Our exploratory analysis indicates that a considerable intratherapeutic increase in ALP may serve as a tool to identify patients who are at a higher risk of early disease progression after PRRT. If our results can be confirmed by other studies, these patients might benefit from intensified follow-up.

**Abstract:**

Background: Pretherapeutic chromogranin A, alkaline phosphatase (ALP), or De Ritis ratio (aspartate aminotransferase/alanine aminotransferase) are prognostic factors in patients with metastatic neuroendocrine tumors (NET) undergoing peptide receptor radionuclide therapy (PRRT). However, their value for intratherapeutic monitoring remains unclear. We evaluated if changes in plasma markers during PRRT can help identify patients with unfavorable outcomes. Methods: A monocentric retrospective analysis of 141 patients with NET undergoing PRRT with [^177^Lu]Lu-DOTATOC was conducted. Changes in laboratory parameters were calculated by dividing the values determined immediately before each cycle of PRRT by the pretherapeutic value. Patients with low vs. high PFS were compared with the Wilcoxon rank-sum test. Results: Progression, relapse, or death after PRRT was observed in 103/141 patients. Patients with low PFS showed a significant relative ALP increase before the third (*p* = 0.014) and fourth (*p* = 0.039) cycles of PRRT. Kaplan–Meier analysis revealed a median PFS of 24.3 months (95% CI, 20.7–27.8 months) in patients with decreasing ALP values (Δ > 10%) during treatment, 12.5 months (95% CI, 9.2–15.8 months) in patients with increasing ALP values (Δ > 10%), and 17.7 months (95% CI, 13.6–21.8 months) with stable ALP values (Δ ± 10%). Conclusions: Based on these exploratory data, a rise in plasma ALP might indicate disease progression and should be interpreted cautiously during therapy.

## 1. Introduction

Peptide receptor radionuclide therapy (PRRT) is a well-established and effective therapy in patients with metastasized gastroenteropancreatic neuroendocrine tumors (GEP-NETs). [^177^Lu]Lu-DOTATATE received approval from the European Medicines Agency (EMA) in 2017 for patients with GEP-NET of low- and intermediate-grade G1-G3 NET who progressed on first-line somatostatin analog (SSA) therapy. In 2018 [^177^Lu]Lu-DOTATATE therapy gained FDA approval for the same indication. Standard therapeutic protocols consist of four cycles administered every 8–10 weeks. Response to treatment can reliably be evaluated 3–6 months after completion of the last treatment cycle. Considering a typical treatment length of 8–12 months and the costs associated therewith, a sufficient response assessment to therapy is indispensable for disease management.

According to current guidelines, disease progression under PRRT is based on morphologic and functional imaging [1,2]. For this purpose, patients receive staging imaging such as somatostatin receptor (SSR)-PET/CT-MRI and subsequent contrast-enhanced (CE)-CT/-MRI 3–6 months after completion of therapy [2]. The selection of the time intervals between the examinations (3–6 months) depends mainly on the grading of the NET [1,2]. However, as described in the current ENETS guidelines, an increase of more than 25% in the tumor marker chromogranin A (CgA) might be suggestive of disease progression and should give cause for CT, MRI, or SSR imaging anytime earlier [1]. Despite this/the ENETS guideline, several recent studies suggest that intratherapeutic changes in the monoanalyte secretory peptide CgA in comparison to baseline show no correlation with response to PRRT [3,4,5]. On the other hand, further blood-based biomarkers like the alkaline phosphatase (ALP) or the De Ritis ratio show evidence of representing the hepatic tumor burden of NET, but the prognostic value of their intratherapeutic alterations has not been evaluated to date [6,7]. Most recently, our research group was able to show that a high pre-therapeutic De Ritis ratio and CgA (>204 µg/L) are independent prognostic factors in patients with NET undergoing PRRT and are significantly associated with a shorter PFS in this patient population [8]. An alternative or supplement to response prediction is imaging-based markers. Heterogeneity parameters and “intensity” of uptake at SSR-PET/CT or intratherapeutic SPECT/CT are associated with morphological treatment response [9,10,11]. However, the simplicity, cost-effectiveness, and high availability of blood-based parameters provide an intriguing alternative to complex and time-consuming semiautomatic imaging methods nowadays. As an example, Arnold et al. have suggested that a sudden increase in CgA could indicate an unfavorable outcome [12].

The aim of this exploratory analysis was to describe intratherapeutic alterations of the De Ritis ratio, ALP, and CgA seen at different cycles of PRRT and to investigate if these alterations in plasma markers could help to predict unfavorable PFS. This analysis did not aim at confirming the results of our previous publication [8] on pretherapeutic prognostic factors, which shared a majority of patients.

## 2. Materials and Methods

### 2.1. Patient Population

This retrospective, single-center study examined 141 patients with histologically confirmed NET who underwent PRRT between September 2007 and August 2021. All patients met previously published inclusion criteria [8], and parts of this study have been previously published [8], but the current analysis includes patients who were treated more recently. Furthermore, the previous study did not evaluate/focus on the usefulness of blood-based biomarkers during intratherapeutic follow-up to detect patients with unfavorable outcomes. 

### 2.2. [^177^Lu]Lu-DOTATOC-PRRT and Response Assessment

PRRT and response assessment were conducted as previously described [8]. Briefly, patients received PRRT, comprising a median of three treatment cycles (range: 2–6 cycles), with each cycle involving a prescribed dosage of 200 mCi (7.40 GBq) of [^177^Lu]Lu-DOTATOC. After two PRRT cycles, patients underwent interim staging with SSR-PET/CT using [^68^Ga]Ga-DOTATOC, typically scheduled every two cycles, with a minimum two-month interval to prevent misinterpretation [13]. Disease progression was determined by an interdisciplinary tumor board, and if observed, no more PRRT cycles were administered. After treatment completion, patients had follow-up imaging every 3 to 6 months. Patients were categorized into groups based on the extent of liver metastases. 

### 2.3. Evaluation

CgA, AST, ALT, and ALP levels were assessed within one week before the administration of each PRRT cycle, and the De Ritis ratio was calculated as AST/ALT. All blood samples were analyzed by the same certified laboratory (Labor Berlin—Charité Vivantes GmbH, Berlin, Germany) with the following reference values:Chromogranin A (CgA): <102 ng/mL;AST: <35 U/L (female), <50 U/L (male);ALT: ≤31 U/L (female), ≤41 U/L (male);ALP: 35–105 U/L (female), 40–130 U/L (male).

PFS was defined as the duration from the initiation of the first PRRT cycle until the identification of progressive disease in accordance with the response evaluation criteria in solid tumors (RECIST 1.1). The cessation of therapy occurred either due to toxicity or as a result of death from any cause [14]. Limited and extensive liver disease was categorized following the criteria outlined by Brabander et al. [13]. Briefly, extensive liver disease was characterized by the presence of massive, centrally necrotizing metastases or miliary metastases involving the entire liver parenchyma, often accompanied by hepatomegaly.

### 2.4. Statistical Analysis

Statistical analysis was performed using SPSS version 26 (IBM, Chicago, IL, USA). Significance was assumed at α = 0.05. Descriptive values were expressed as median and range. The Kaplan–Meier method was used to estimate survival rates and median PFS. To investigate the association between changes in laboratory parameters with PFS, patients were separated into groups of low vs. high PFS. The PFS cut-off was defined as the median value in the total patient sample (*n* = 141). Patients without disease progression/relapse who had a follow-up duration less than this PFS cut-off were excluded from further analysis. Relative values for CgA, AST, ALT, ALP, and De Ritis ratio were calculated for each cycle by dividing the values determined immediately before each cycle of PRRT by the value determined immediately before the application of the first cycle of PRRT. The distribution of relative values between the two groups of low vs. high PFS was compared for the first four cycles of PRRT using the Wilcoxon rank-sum test and was visualized by plotting the medians and their 95% confidence intervals (95% CI). Categorical variables were compared between the same two groups with Fisher’s exact test. Furthermore, changes in ALP during PRRT were compared between patients with vs. without osseous metastases or with extensive vs. limited liver metastases using the Wilcoxon rank-sum test. To further examine the prognostic relevance of relative ALP changes during PRRT, patients in the study group (*n* = 121) were separated based on the magnitude of relative ALP change from the individual last cycle of PRRT (i.e., 4th cycle or less) compared to the pretherapeutic ALP. An ALP increase or decrease of >10% was defined as an increase/decrease, while patients with ALP changes within ±10% of the pretherapeutic ALP were defined as stable. ALP changes at PRRT cycles other than the first or last cycle were not considered for this analysis.

## 3. Results

### 3.1. Patients

While 67/141 patients (48%) suffered from a primary tumor located in the small intestine, 38/141 patients (27%) had a pancreatic primary, 12/141 patients (9%) showed a primary tumor located in the colon or rectum, 10/141 patients (7%) suffered from a pulmonary primary, 1/141 patient (1%) had a primary tumor located in the stomach, and 13/141 patients (9%) suffered from cancer of unknown primary (CUP). In 136 of 141 patients (97%), PRRT followed previous treatments (operative resection, *n* = 93; somatostatin analogs, *n* = 108; mTOR inhibitor, *n* = 17; tyrosine kinase inhibitor, *n* = 4; chemotherapy, *n* = 41; local ablative therapy, *n* = 12; radiation therapy, *n* = 5; transcatheter arterial chemoembolization, *n* = 7). Table 1 illustrates all patient characteristics. 

### 3.2. Progression-Free Survival

In total, 103 out of 141 patients (73%) experienced disease progression. The median PFS for the complete cohort of 141 patients was 19.5 months (95% CI 15.7–23.3 months), as illustrated in Figure 1. Patients who did not experience disease progression or relapse had a median follow-up duration of 18.3 months, ranging from 5.5 to 43.9 months. Consequently, 20 patients were excluded from the groups categorized by high and low PFS because their follow-up duration was shorter than the PFS cut-off (details in Appendix A). Accordingly, the study cohort comprised 121 patients. Notably, no cases of nephrotoxicity grade ≥3, hematologic toxicity grade ≥3, tumor lysis syndrome, or dose-limiting liver damage were observed.

### 3.3. Changes in Laboratory Parameters

Immediately before the application of the first cycle of PRRT, the median CgA value of the study cohort was 452 (range: 14–601,700). In the period between the first and second cycle and between the second and third cycle, CgA slightly decreased and was 427 (range: 16–560,300) before the second cycle and 319 (range: 27–419,300) before the third cycle. A discrete increase to a median of 324 (range: 26–259,100) occurred between the third and fourth cycles. Detailed information for CgA, AST, ALT, De Ritis ratio, and ALP at each PRRT cycle is shown in Table 2.

### 3.4. Alterations of Laboratory Parameters in Relation to the First Cycle PRRT

Table 3 presents the relative changes in the laboratory values to the baseline values before the first cycle of PRRT.

Figure 2A–C show the medians of relative laboratory values and their 95% CI separated by the two patient groups with low vs. high PFS for each PRRT cycle. The histogram distribution of ALP variation is illustrated in Appendix A. In the study cohort (*n* = 121), no significant differences were observed between patients with or without osseous metastases with regard to the relative ALP values before the second (*p* = 0.29), third (*p* = 0.59), and fourth cycle (*p* = 0.64). We observed significantly higher baseline ALP levels in patients with extensive (median: 75 U/L; range: 36–296 U/L) vs. limited liver metastases (median: 102 U/L; range: 52–470 U/L; *p* < 0.001). However, we did not find significant differences between patients with extensive vs. limited liver disease in the ALP ratio from the third cycle to the first cycle (1.04 [0.57–1.45] vs. 0.98 [0.41–1.4]; *p* = 0.237) or the fourth cycle to the first cycle (1.12 [0.71–1.77] vs. 1.02 [0.44–1.48]; *p* = 0.582).

### 3.5. Prediction of PFS

In the study cohort (*n* = 121), Kaplan–Meier analysis revealed a median PFS of 24.3 months (95% CI, 20.7–27.8 months) in patients with decreasing ALP values (Δ > 10%) during treatment compared to 12.5 months (95% CI, 9.2–15.8 months) in patients with increasing ALP values (Δ > 10%). Patients with stable ALP values (Δ ± 10%) showed a median PFS of 17.7 months (95% CI, 13.6–21.8 months; log-rank test, *p* = 0.142; Figure 3).

## 4. Discussion

Investigating pretherapeutic laboratory parameters along with their intratherapeutic alterations of patients scheduled for PRRT, the change in ALP before administration of the third and the fourth PRRT cycle showed a significant association with shortened PFS. In addition, ALP distribution between the two subgroups (≤19.5 months vs. >19.5 months) directly obtained before the second cycle PRRT displays a trend that might also help identify “high-risk” patients who will experience rapid progression, although the results were not significant. Prevalence of elevated intratherapeutic CgA as well as De Ritis ratio, however, had no relevant impact on PFS. 

A reversible initial increase in CgA, ALP, AST, and ALT is a well-known phenomenon after administration of PRRT cycles, and in the vast majority of patients, this will resolve during follow-up [13]. As a possible explanation, liver enzymes and CgA are suspected to increase via radiation therapy-induced liver disease, which typically presents after 4–8 weeks of initial treatment administration [15]. Clinicians should be aware of these changes in blood-based parameters, highlighting this as a potential pitfall. Nevertheless, changes may also occur as a first sign of disease progression. The key challenge therefore is to clearly distinguish between pseudoprogression and tumor progression.

In a previous study, we were able to demonstrate that elevated De Ritis ratio (>0.927) as well as elevated CgA (>204 µg/L) prior to PRRT are independent prognostic factors in patients with NET and are significantly associated with a shorter PFS in this patient group [8]. With respect to other cancer types, higher values of ALP are significantly associated with poorer outcomes in patients with osteosarcoma, breast cancer, and prostate cancer [16,17,18]. Additionally, recent studies suggest that ALP could be a prognostic parameter in patients with liver metastases from NET undergoing transarterial chemoembolization [6]. We hypothesize that ALP changes during therapy could be indicative of an increase in hepatic tumor burden and thus of progressive disease. In addition, increased ALP may also indicate an increased level of bone remodeling and could highlight the presence of osseous metastases [13]. Nevertheless, in the current patient sample, pretherapeutic ALP did not differ between patients with vs. without osseous metastases, and neither did ALP changes during PRRT. Additionally, in a recent post hoc analysis of the prospective NETTER-1 cohort, baseline ALP did not affect the survival outcome after PRRT [19]. We must emphasize that the present work is an exploratory analysis and may be confounded via multiple statistical tests. 

To the best of our knowledge, this is the first study to evaluate the usefulness of intratherapeutic alterations of the De Ritis ratio to identify patients with NET that show unfavorable outcomes after PRRT, although the current analysis is explorative. Various explanations exist for the pretherapeutic prognostic significance of the De Ritis ratio in different cancer types. The leading explanation suggests a relationship between the De Ritis ratio and an increased level of anaerobic glycolysis in cancer cells, called the “Warburg effect” [8,20,21,22,23,24,25,26,27,28]. In this context, there are various interactions between an altered NADH/NAD+ ratio and the malate– aspartate shuttle, which is involved in the NADH supply of mitochondria and for whose function AST is essential [27,29,30,31,32]. In addition, Thornburg et al. demonstrated that cancer cells depend on AST to show high proliferation rates [33]. Thus, an elevated De Ritis ratio could be indicative of an increased proliferation rate of tumor cells. Since the proliferation rate does not differ in the short term, this could explain why the De Ritis quotient was approximately constant between the cycles and between the patient groups divided according to the PFS. However, this explanation remains speculative.

The results obtained in the present study are well in line with those of Bodei et al. and Brabander et al. [3,13]. Bodei et al. evaluated the levels of CgA just before administration of PRRT cycles and their predictive power on treatment/tumor response, which was determined via initial post-therapeutic imaging [3]. The authors found that CgA alterations did not differ between RECIST responders and non-responders [3]. However, it should be emphasized that Bodei et al. analyzed absolute values of CgA, and statistical inaccuracies may have occurred since CgA values usually differ widely between patients [3]. Overall, it remains unclear why intratherapeutic alterations of CgA, in contrast to the absolute pretherapeutic values, seem to be unrelated to the treatment outcome. It should be noted that CgA is a parameter that is highly susceptible to interference [34]. For example, Fossmark et al. demonstrated that the use of proton-pump inhibitors regularly leads to an increase in CgA [34,35]. Many other clinical conditions, such as kidney failure, liver cirrhosis, arterial hypertension, or chronic heart failure, can also cause increased CgA values [34,36,37,38]. Changes in CgA should therefore only be considered with extreme caution and should not be used to assess response to therapy.

This study had several limitations, including its retrospective, exploratory nature and the absence of a matched control group of patients undergoing a different treatment modality. Validation with a larger, independent dataset is required to draw definitive conclusions. Furthermore, prospective studies would be needed to ensure a well-defined and homogeneous patient cohort with a uniform follow-up.

## 5. Conclusions

In patients with NET scheduled for PRRT, a considerable increase in plasma ALP was associated with shortened PFS, while intratherapeutic alterations of CgA and De Ritis ratio did not correlate with PFS. A rise in plasma ALP might be a potential indicator of disease progression and should be interpreted cautiously during therapy. However, these results are explorative and need to be further validated.

## Figures and Tables

**Figure 1 cancers-15-05717-f001:**
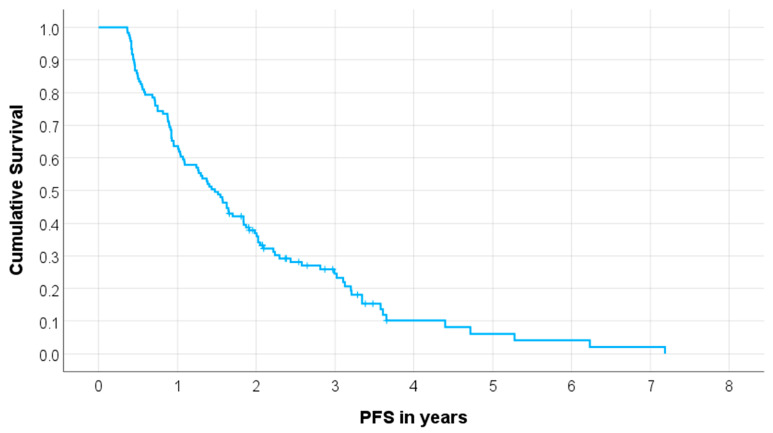
Kaplan–Meier curve for progression-free survival (PFS) in the total cohort (*n* = 141).

**Figure 2 cancers-15-05717-f002:**
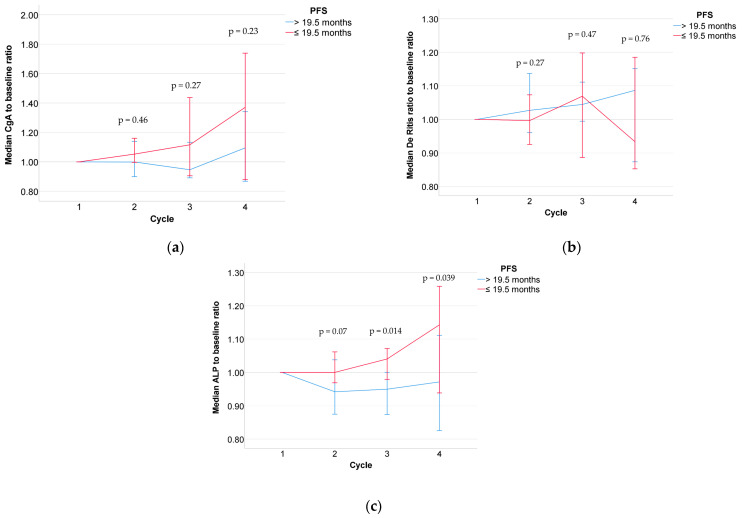
Medians of relative laboratory values CgA (**a**), De Ritis ratios (**b**), and ALP (**c**) with their 95% confidence intervals at each PRRT cycle separated by median PFS.

**Figure 3 cancers-15-05717-f003:**
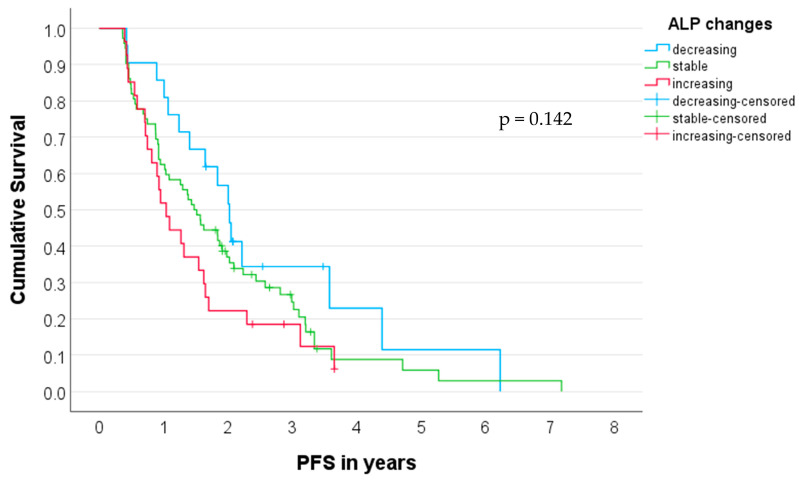
Kaplan–Meier plot for progression-free survival (PFS) in the study cohort (*n* = 121) for patients with increasing, stable, and decreasing ALP values.

**Table 1 cancers-15-05717-t001:** Patient characteristics.

Variable	*n* (%) or Median (Range)			*p*
	Total Cohort	PFS Low	PFS High	
Patient count	141	67	54	
Age in years	64 (34–87)	66 (42–87)	63 (34–80)	0.36
Sex				1.00
Men	89 (63%)	44 (66%)	36 (67%)	
Women	52 (37%)	23 (34%)	18 (33%)	
Primary location				**0.009**
Small intestine	67 (48%)	29 (43%)	26 (48%)	
Pancreas	38 (27%)	25 (37%)	11 (20%)	
Colon/Rectum	12 (9%)	2 (3%)	8 (15%)	
Lungs	10 (7%)	7 (10%)	1 (2%)	
Stomach	1 (1%)	0 (0%)	1 (2%)	
CUP	13 (9%)	4 (6%)	7 (13%)	
Metastatic disease	138 (98%)	66 (99%)	53 (98%)	1.00
Metastatic spread				
Hepatic	135 (96%)	64 (96%)	52 (96%)	1.00
Lymphonodal	118 (84%)	59 (88%)	43 (80%)	0.22
Osseous	54 (38%)	29 (43%)	18 (33%)	0.35
Peritoneal	24 (17%)	11 (16%)	9 (17%)	1.00
Pulmonal	6 (4%)	3 (5%)	3 (6%)	1.00
Functional tumor	46 (33%)	19 (28%)	19 (35%)	0.44
Hedinger syndrome	6 (4%)	3 (5%)	1 (2%)	0.63
Grading				0.98
G1	30 (21%)	12 (18%)	11 (20%)	
G2	100 (71%)	50 (75%)	39 (72%)	
G3	5 (4%)	3 (5%)	2 (4%)	
Unknown	6 (4%)	2 (3%)	2 (4%)	
Ki-67 index	5 (1–40)	10 (1–40)	5 (1–25)	0.10
Number of PRRT cycles	3 (2–6)	3 (2–6)	4 (2–6)	**0.001**
Previous treatment				
Operative resection	93 (66%)	44 (66%)	36 (67%)	1.00
Somatostatin analogues	108 (77%)	51 (76%)	41 (76%)	1.00
mTOR inhibitor	17 (12%)	11 (16%)	6 (11%)	0.44
Tyrosine kinase inhibitor	4 (3%)	3 (5%)	1 (2%)	0.63
Chemotherapy	41 (29%)	25 (37%)	14 (26%)	0.24
Local ablative therapy	12 (9%)	6 (9%)	5 (9%)	1.00
Radiation therapy	5 (4%)	3 (5%)	2 (4%)	1.00
Transcatheter arterial chemoembolization	7 (5%)	4 (6%)	1 (2%)	0.38

Patient characteristics are provided for the total cohort and separated for patients with low or high PFS, respectively. Both subgroups were compared using Fisher’s exact test or Wilcoxon rank-sum test.

**Table 2 cancers-15-05717-t002:** Values of laboratory parameters for each PRRT cycle (*n* = 141 patients).

Laboratory Parameter	Median (Range)			
	1st Cycle	2nd Cycle	3rd Cycle	4th Cycle
CgA (in µg/L)	452 (14–601,700)	427 (16–560,300)	319 (27–419,300)	324 (26–259,100)
AST (in U/L)	29 (13–123)	28 (14–104)	27 (15–84)	27 (15–75)
ALT (in U/L)	27 (10–122)	25 (7–215)	24 (9–95)	25 (14–97)
De Ritis ratio	1.11 (0.52–2.87)	1.09 (0.28–2.86)	1.15 (0.38–3.38)	1.05 (0.45–3.06)
ALP (in U/L)	84 (36–470)	81 (36–385)	74 (34–263)	77 (42–272)

Laboratory parameters were determined < 1 week before application of each cycle.

**Table 3 cancers-15-05717-t003:** Relative values of laboratory parameters (*n* = 121 patients).

Laboratory Parameter	Median (Range)			*p*
	Total Cohort	PFS Low	PFS High	
CgA				
2nd Cycle/1st Cycle	1.06 (0.19–3.73)	1.05 (0.24–2.88)	1.00 (0.19–3.73)	0.46
3rd Cycle/1st Cycle	1.07 (0.08–3.76)	1.12 (0.24–3.76)	0.95 (0.08–3.22)	0.27
4th Cycle/1st Cycle	1.15 (0.12–3.82)	1.37 (0.24–3.69)	1.10 (0.12–3.82)	0.23
AST				
2nd Cycle/1st Cycle	0.96 (0.26–2.31)	0.96 (0.26–2.31)	0.94 (0.48–2.02)	0.75
3rd Cycle/1st Cycle	0.98 (0.45–2.12)	0.99 (0.45–2.05)	1.00 (0.57–2.12)	0.84
4th Cycle/1st Cycle	1.00 (0.47–2.21)	0.96 (0.65–1.46)	0.98 (0.47–2.21)	0.73
ALT				
2nd Cycle/1st Cycle	0.84 (0.16–4.78)	0.82 (0.16–2.43)	0.87 (0.35–4.78)	0.42
3rd Cycle/1st Cycle	0.84 (0.26–2.65)	0.94 (0.26–2.00)	0.83 (0.46–2.65)	0.74
4th Cycle/1st Cycle	0.92 (0.39–2.88)	0.89 (0.46–1.82)	0.94 (0.39–2.88)	0.47
De Ritis ratio				
2nd Cycle/1st Cycle	0.99 (0.39–1.88)	1.00 (0.48–1.88)	1.03 (0.53–1.84)	0.27
3rd Cycle/1st Cycle	1.04 (0.40–2.16)	1.07 (0.40–1.86)	1.04 (0.64–2.16)	0.47
4th Cycle/1st Cycle	1.03 (0.61–1.67)	0.93 (0.65–1.61)	1.09 (0.67–1.40)	0.76
ALP				
2nd Cycle/1st Cycle	0.99 (0.51–2.95)	1.00 (0.61–2.95)	0.94 (0.51–2.46)	0.07
3rd Cycle/1st Cycle	1.00 (0.41–1.45)	1.04 (0.57–1.45)	0.95 (0.41–1.40)	**0.014**
4th Cycle/1st Cycle	1.03 (0.44–1.77)	1.14 (0.73–1.77)	0.97 (0.44–1.53)	**0.039**

The relative value was calculated by dividing the value determined immediately before each cycle by the value determined immediately before the application of the first cycle of PRRT. Values are provided for the study cohort (*n* = 121) and separated for patients with low or high PFS (>19.5 months), respectively. Both subgroups were compared using the Wilcoxon rank-sum test.

## Data Availability

The data presented in this study are available on request from the corresponding author. The data are not publicly available due to ethical restrictions.

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
