# Peer review of "Plasma Markers for Therapy Response Monitoring in Patients with Neuroendocrine Tumors Undergoing Peptide Receptor Radionuclide Therapy"

_cancers, 2023, doi:10.3390/cancers15245717_

Round 1

Reviewer 1 Report

Comments and Suggestions for Authors

In general, the work is interesting and has elements for clinical use in radioligand therapy - RLT, formerly PRRT, which is used by the authors in their study.

A few "technical" comments on the presentation of data in tables and in general on the statistical methods used and their selection without relying on the Cox proportional hazards regression model, after taking into account more variables in the moedl built.

The size of the study group allows the construction of a reliable moedl.

Below are some comments and remarks to the authors.

In Table 3 line 185 page 6

The results presented CgA ratio in subsequent therapies in relation to the initial CgA measurement before therapy has an increasing trend, from the text it appears that it has a decreasing trend subsequent CgA determination during subsequent courses of treatment, decreases, please verify the data in Table 3;

Similar comments apply to ALP levels, where in the text the authors show a favorable prognosis for patients with a decrease in ALP, seen only slightly in the high PFS group, and this is most clear between the 1st and 2nd doses of PRRT, and virtually similar between the 1st and 4th doses.

Why authors used Wilcoxon rank-sum test, instead Wilcoxon signed-rank test, which seems to be more useful in this case. The test is used to compare two related samples, matched samples as presented by the authors data sets;

Discussion

The assumption made by the authors about significantly lowering ALP between 1 and 3 and 1 and 4 courses of RLT (PRRT) treatment combined with high PFS vs low PFS, may be an oversimplification in the model built, other prognostic factors affecting the predictive assessment of response and prognostic assessment such as NETG2 vs NETG1, pancreas vs bowel etc. should be considered more;

Further factors that should be considered in the model are for example: clinical stage, the degree of liver involvement, presence of boe mts only by separating the described groups can one be tempted to make comparisons such as those described by the authors.

In conclusion, the authors unfortunately took a bit of a shortcut by presenting a one-sided view of the assessment of response to RLT (PRRT) therapy, grouping patients with low PFS below the median and high PFS above the median solely on a few selected elements of the bochemical analysis without building and checking the moedle on the basis of statistical analysis based on the Cox proportional hazards regression model and only after assessing the significance of the parameters included in the analysis considering them as potential predictive, prognostic factors.

Author Response

Dear Reviewer,

We appreciate the opportunity to submit our revised manuscript titled "Plasma Markers for Therapy Response Monitoring in Patients with NET Undergoing PRRT" and thank you for your positive and constructive comments. We have provided point-by-point responses to your comments and have highlighted the modifications to the revised manuscript in red.

Please find the Reply to the Reviewers' comments attached as a Word document.

Kind regards, Christoph Wetz

Reviewer 2 Report

Comments and Suggestions for Authors

The article is well-written and presented. However, I would change the article's title to something else: the fact that elevated LFTs are predictive of a shorter PFS is not a novel concept or biomarker. This is almost certainly indicative of the progression of liver disease/dysfunction and is not a new concept. 

Author Response

(The authors gave the same response as above.)

Reviewer 3 Report

Comments and Suggestions for Authors

Christoph et al. found that an intratherapeutic increase of ALP may serve as an effective prognostic tool for therapy response monitoring in patients with GEP-NET undergoing PRRT. These are some issues that need to be addressed.

1.     There were only 67 patients in PFS low group and 54 in PFS high group. However, there are 141 patients in total cohort. Specific information for the 20 excluded patients should also be added to the table.

2.     Why 20 patients without disease progression/relapse who had a follow-up duration less than this PFS cut-off were excluded from further analysis.

3.     Whether the CgA, AST, ALT, and ALP were tested in the same medical center.

4.     The main limitation of the study is its small sample size with only 121 patients included in the study cohort for further analysis, the study is not powered for the analyses described. Increased sample size and external validation are strongly recommended.

Author Response

(The authors gave the same response as above.)

Reviewer 4 Report

Comments and Suggestions for Authors

 Authors have already published in 2021 a retropective analysis in 125 patients with various NET locations all treated with PRRT showing that a high pre-  therapeutic De Ritis ratio and CgA (> 204µg/l) are independent prognostic factors in NET patients  undergoing PRRT as associated with PFS irrespective of the liver tumor burden.

 This time they present a second study intially presented as to be on more patients, 141 patients with an analyssis in which ALP is added

The PFS cut- off was defined as the median value in the total patient sample (n=141)- why the median was chosen as PFS cut off ?

Patients without disease progression/relapse who had a follow-up duration less than this PFS cut-off were  excluded from further analysis. Therefeore  121 patients consisted the study group.

Finally data analysis was perfomerd in 121 patients and not in 141.

This gets me really confused becouse in the previous paper the analysis was conducted in 125 patients whether in the present study the analysis was conducted in 121 patients

 As ALP varied in adifferent manner in patients with bone métastases, why were not these patient analyzed separately ?Although no statistic differences were registered between bone and no bon metastases groups, the SD for partients with bone metastases remain important, which would explain the lack of statistical difference between groups. However , it would be interesting to analyze patients with bone metastases separately and try to see whethere there are any correlations between bone tumoral volume (although very difficult to assess in NET patients as very few measurable lesions) and alkaline phosphatase variations, in responders versus non responders.

It would be easier to follow up if you could add the p values for each histogram from Figures 2 and 3.

 Although in the previous study authors found that elevated De Ritis ratio as well as elevated CgA prior to PRRT are independent prognostic factors in patients with NET ignificantly associated with a shorter PFS in the present study this correlation is not confirmed.

 First , the paragraph  lines 245-247 is no longer true as authors have already published a manuscript realted to the subject.

Authors try to get an explanation to the lack of correlation of De Ritis ratio to PFS in the present study , but this explanation remains to debate.

The 121 patients analyzed in this study in majority should be the same as in the previously published study, I presume  as retrospective from 2007 to 2021 and theprevious published manusrcipt included patients from 2007 to 2019 at least.

 In this case your supposition of absence of rapid progression should be the same for the two groups ; however in the previous anaysis a difference was noticed

Could you please explain in a statistic or mathematical way the statstically differences observed in this manuscript compared to the previous one ?

As the authors underline as limit of the study is that in order to confirm the value of the Ritis ratio  to predict outcome , this shouls be validated in an independent cohort, which, as I presume was not the case.

Authors should highlight that the objective of this manuscript was to analyse the value of ALP  in order to predict outcome as PFS.

A second objective was to check whether they confirm in  the augmented previous cohort the published results, which was finally not the case.

Finally which were the differences between the two cohorts , in grade or Ki67 that could explain that less rapid proegression was noticed in the present cohort compared to the previous one?

Comments on the Quality of English Language

No comments

Author Response

(The authors gave the same response as above.)

Round 2

Reviewer 1 Report

Comments and Suggestions for Authors

Dear Authors;

As for the authors' answers to the questions asked about the tabulated values of CgA and ALP, perhaps more readable for the readers, would be the use of Upper Limit Normal (ULN) multiples in the assessment of potential changes during the course of therapy, particularly useful in the assessment of ALP, less so in the case of CgA, because the values of CgA are of a very wide range from low values <ULN to extremely high.

Slightly better is the possibility of presenting ALP results here the range is not with such a large range of results and for readability one can make ULN calculations as more useful then it may turn out that the distribution will be more different which will be associated with easier interpretation of data.

An important part of this work should be to present the results in such a way that the conclusions suggested by the authors as to the role of ALP in predicting further progression already on therapy can be routinely applied.

Therefore, the data presented by the authors given in the table are difficult to read as to their real value and bring out the essence of the work.

The presented graphs of Fig 2. pertaining to the values of CgA, De Rits and ALP should be presented as columns, starting with a comparison of 2 therapies to 1; 3 to 1, etc. and then the graph data will be more readable, for CgA and ALP; instead of CI+/- 95%), I propose to use SEM/SD;

Fig 2 D, is not integrally related to the concept of Fig. 2 and should be removed.

Fig 1 proposes to present time in months 12, 24, 36, 48 months etc. Easier to read years of follow-up, on the ordinate axis of Cumulative Survival I propose to put scales in increments of 0.1 so as to have 0.5 i.e. median legibly on the graph;

Similarly for Fig 3.

Answer 3 is satisfactory;

The clarification in regarding the discussion along with the changes in the abstract clean up the paper and its purpose.

However, there is still an untapped potential of the work with 141 patients and 121 patients analyzed with sufficient follow-up documented with determination of PFS, full data should be developed by the authors in a full data sets analysis.

In addition, it would be useful to show potential differences in ALP based on the assessment in PFS in patients with massive, mediocre or no mts to the liver.

Author Response

Dear Reviewer,

Thank you once again for your thorough review and valuable feedback on our manuscript. We have carefully considered your comments and appreciate your recognition of our efforts to address them to the best of our ability, even though we may have differing opinions on the first two points. Nevertheless, your input has been invaluable in enhancing the quality and clarity of our manuscript. Please find a Word document with all comments attached to this message.

Kind regards,
Christoph Wetz

Reviewer 3 Report

Comments and Suggestions for Authors

After review of your revised manuscript and review of your responses, all comments were answered appropriately and the manuscript improved

Author Response

Dear Reviewer,

Thank you for your feedback and for reviewing our revised manuscript! We appreciate your thorough review and are pleased to hear that all your comments have been addressed appropriately.

Kind regards,

Christoph Wetz

Reviewer 4 Report

Comments and Suggestions for Authors

Authors provided clear precisions of their aims and methods and thorough responses

Authors provided thorough revision and good commentes and a revised version of the mansucript. They clearly explain the lack of correlation of CgA ratio versus baseline  to disease evolution and PFS, as well as the lack of De Ritis ratio correlation with PFS

 The variation of ALP during treatment is interesting and deserves further exploration and validation, especially that ALP is produced mainly in liver , but also in bone and intestine,and from th analitical point  of view it is difficult to correctly separate the ALP fractions of liver rather than bone or intestinal origin. If there was no difference in ALP variation with bone metastasis status, it would be interesting to look at differences in liver metastasis status.

 Authors suggest that "We hypothesize that ALP changes during therapy could be indicative of an increase in hepatic tumor burden and thus of progressive disease" as previously ALP was published to be predictive of liver metastasis status and response to chemoembolization.

It would be important that authors add in Figure 2 a histogram and analysis related to ALP variation between cycles in low vs high PFS groups, especially that liver metastasis distribution between the two groups seems not different. 

Comments on the Quality of English Language

Good quality of English

Author Response

Dear Reviewer,

Thank you once again for your thorough review and valuable feedback on our manuscript! We appreciate your acknowledgment of our efforts to address your comments to the best of our ability. Your input has been invaluable in enhancing the quality and clarity of our manuscript. Please find a Word document with all comments attached to this massage. 

Kind regards,

Christoph Wetz
